

# A splash in a one-dimensional cold gas

Subhadip Chakraborti[1], Abhishek Dhar[1] and P. L. Krapivsky[2,3]

**1** International Centre for Theoretical Sciences, Tata Institute of Fundamental Research,
Bengaluru 560089, India
**2** Department of Physics, Boston University, Boston, MA 02215, USA
**3** Santa Fe Institute, 1399 Hyde Park Road, Santa Fe, NM 87501, USA

## Abstract

We consider a set of hard point particles distributed uniformly with a specified density on the positive half-line and all initially at rest. The particle masses alternate between two values, $m$ and $M$. The particles interact via collisions that conserve energy and momentum. We study the cascade of activity that results when the left-most particle is given a positive velocity. At long times we find that this leads to two fascinating features in the observed dynamics. First, in the bulk of the gas, a shock front develops separating the cold gas from a thermalized region. The shock-front travels sub-ballistically, with the bulk described by self-similar solutions of Euler hydrodynamics. Second, there is a splash region formed by the recoiled particles which move ballistically with negative velocities. The splash region is completely non-hydrodynamic and we propose two conjectures for the long time particle dynamics in this region. We provide a detailed analytic understanding of these coexisting regimes. These are supported by the results of molecular dynamics simulations.


doi:10.21468/SciPostPhys.13.3.074

# 1 Introduction

Consider the infinite-space version of the billiard or carrom board problem: hard discs are initially at rest and uniformly distributed in the half-space $x \geq 0$, and the system is perturbed by kicking a particle near the origin in the $x$ direction. The moving particle eventually hits a particle at rest creating further collisions and generating a cascade penetrating the occupied half-space. Particles are also ejected back and a splash-like pattern is formed [see Fig. (1)]. This splash problem can be thought of as a billiard with no walls and an infinite number of particles. Despite a simple formulation, the splash problem greatly differs from traditionally studied billiards systems [1,2]. The splash problem was mentioned in [3,4]. Here we provide the first detailed treatment for the one-dimensional version of the problem.

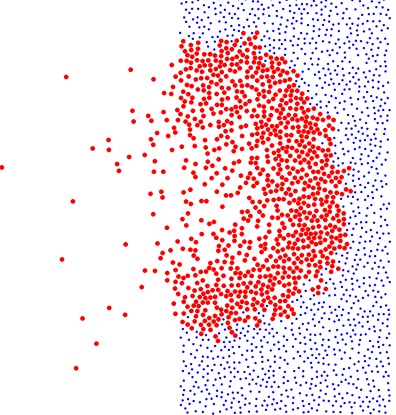

Figure 1: A splash in two dimensions: this shows a snapshot of a gas of hard discs, initially at rest and uniformly distributed on the positive half plane, some time after one disc near the origin is given a velocity in the positive $x$–direction. Moving particles are shown by red disks; stationary ones are shown by blue disks and their size is shrunk for visual convenience (the figure is adopted from Ref. [3]).

In one dimension collisions are inevitable, so it suffices to consider point particles. The point particles make the system infinitely diluted and hence ideal with the well-known equation of state. The case with equal masses is pathological — each collision leads to the exchange of identities, with no relaxation. To avoid pathology, we consider a hard point gas with binary mass distribution and assume that particles with dissimilar masses alternate. This system is non-integrable, has good thermalization behavior, and continues to have an ideal gas equation of state. The alternating hard particle (AHP) gas has been extensively investigated in the context of studies on heat transport and hydrodynamics in one dimension [5–15]. In recent work [16,17], the AHP model was used for a numerical verification of the equivalence of the continuum hydrodynamics description and the Newtonian description in the context of the so-called blast problem. The blast problem is similar to the splash problem with the important difference that energy is injected at the centre of a cold gas. The long time behavior of the blast is described by hydrodynamics. The work in [16,17] find that the bulk of the excited gas is described by the famous Taylor-von Neumann-Sedov (TvNS) self-similar solution of the Euler equations, while Navier-Stokes corrections become important in the core region. The splash problem is less tractable than the blast problem due to the lack of symmetry.

Schematically our one-dimensional set-up of the splash problem is the following: we consider the AHP gas of particles initially at rest and uniformly distributed on the positive half-line

$x > 0$. At the time $t = 0$, the left-most particle is given a unit positive velocity. At long times we find that the system evolves to an intriguing state with a hydrodynamic bulk phase coexisting with a non-hydrodynamic "splash" phase [see Fig. (2)]. The splash consists of particles moving ballistically to the left while the hydrodynamic region grows as

$$R(t) = \alpha t^{\delta}, \tag{1}$$

where the exponent $\delta = 0.6279520544\ldots$ is computed analytically; it is smaller than the exponent $\frac{2}{3}$ characterizing the growth of the excited region in the blast problem, equivalently the position of the shock waves on the left and right.

The new feature in the splash problem, as compared to the blast problem, is the existence of this non-hydrodynamic splatter which significantly alters the form of the hydrodynamic scaling, resulting in a self-similar solution of the second kind. Remarkably, we find that a number of particles at the left end of the splash have their velocities frozen, the number of such particles growing with time. Another surprising feature is that asymptotically all energy is in the initially empty half-line $x < 0$. More precisely, the energy of particles in the $x > 0$ half-line decays algebraically as $t^{-(2-3\delta)}$.

The problem that we treat here falls in the broad class of phenomena where self-similar scaling solutions appear in various dynamical systems, but where exponents do not get fixed by dimensional arguments. This leads to anomalous dimensions and these have been referred to as scaling solutions of the second kind [18–20]. Such a solution was first discovered by Guderley [21] who found that the spherical shock wave disappearing at time $t = 0$ shrinks as $(-t)^b$ with exponent $b$ not fixed by dimensional analysis. This problem (and its deformations) is still explored [22,23]. Other self-similar solutions of the second kind have been explored for a wide class of physical phenomena [24–30]. Their importance in physics is also clear [31]. In the present work we present a simple example where we start with a microscopic model and the hydrodynamics treatment naturally leads us to consider solutions of the second kind. In our example we are able to make a direct comparison between results obtained from microscopic simulations and from hydrodynamics. The splash consists of a hydrodynamic and a non-hydrodynamic region (what we call the splatter) and these couple to each other. Importantly the scaling solution in the hydrodynamic region enables us to make predictions for the splatter.

The rest of the paper is organized as follows. In Sec. (2), we define the precise model and present heuristic arguments which allow us to understand some of the important features of the emergent dynamics. In Sec. (3) we present the hydrodynamic theory and the method of determination of the self-similar solution. Numerical results from the microscopic simulations of the hard-particle gas and their comparison with the analytic results are contained in Sec. (4). In Sec. (5) we discuss results for the particles contained in the splatter. We summarize our main findings in Sec. (6).

## 2 Microscopic model and heuristics

We consider particles labeled as $j = 0, 1, 2, \ldots, \infty$ with positions $\{q_j\}$, velocities $\{v_j\}$ and masses $\{m_i\}$. At time $t = 0$, the positions are chosen from a uniform distribution with $0 < q_0 < q_1 < q_2 \ldots$, with mean inter-particle distance $1/n$, and velocities $v_j = v_0 \delta_{j,0}$ ($v_0 > 0$). There is no interaction potential between the particles except for the infinite contact interaction. Thus the particles move ballistically between collisions and can undergo collisions with their nearest neighbors. The collisions conserve energy and momentum so that particles $j$ and

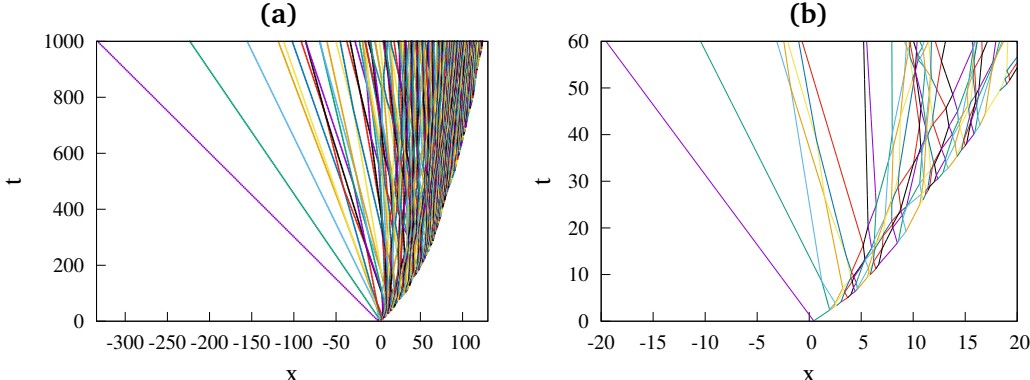

Figure 2: The plot of the space-time trajectories of moving particles of a typical realization of a splash in a one-dimensional gas at (a) long and (b) short times. Notice that at the left end we see a fan of particles moving ballistically, while the position of the right-most moving particle (the shock position) evolves sub-ballistically. In (b) we zoom in to see that the left-most particle undergoes a single collision, the next particle 4 collisions and the third particle 6 collisions before their velocities get frozen.

$j + 1$ have post-collisional velocities

$$v'_j = \frac{(m_j - m_{j+1})v_j + 2m_{j+1}v_{j+1}}{m_j + m_{j+1}} \,, \tag{2a}$$

$$v'_{j+1} = \frac{2m_j v_j - (m_j - m_{j+1})v_{j+1}}{m_j + m_{j+1}} \,. \tag{2b}$$

In the alternating mass setting, the light particles of mass $m$ and heavy particles of mass $M$ can occur in one of the two possible arrangements

$$m_0, m_1, m_2, \ldots = mMmMmMmM \ldots \tag{3a}$$

$$= MmMmMmMm \ldots, \tag{3b}$$

with $v_0$ as the initial velocity of the left-most particle, the entire energy of the system is $\mathcal{E}_0 = mv_0^2/2$ for the first arrangement and $\mathcal{E}_0 = Mv_0^2/2$ for the second. Without loss of generality, we set $v_0 = 1$; the dynamics for any other value of $v_0$ could be obtained by a simple scaling of time and velocities.

With time, an increasing number of particles become active (moving). We begin by providing a heuristic discussion of how the initial energy input is dispersed in the system. A typical evolution showing the space-time trajectories of the active particles is shown in Fig. (2). We see that a small number of splatter particles that are ejected backward suffer a small number of collisions and move ballistically while the major fraction of particles on the right undergo repeated collisions and are confined within a sub-ballistic front. The emerging behavior is in fact quite striking — in the long-time limit $t \gg n^{-1}$, a tiny fraction of the energy is transmitted to the right while the splatter particles carry most of the energy. More precisely, we now argue that the energy $\mathcal{E}(t)$ in the occupied half-line $x > 0$ decays in an algebraic fashion

$$\mathcal{E} \sim t^{-\beta} \,, \tag{4}$$

with $\beta > 0$. This can be seen through the following heuristic arguments: let $v$ be a typical velocity in the cascade. It also gives a typical velocity with which particles are ejected. Therefore

the energy $\mathcal{E}$ in the cascade decays as

$$\frac{d\mathcal{E}}{dt} \sim -mv^2 \times nv \,, \tag{5}$$

since $mv^2$ is the typical energy of ejected particles and $nv$ ($n$ being the mean number density of particles) is the rate at which particles enter the initially empty half-line $x < 0$. For a scaling solution, the position of the shock front can only depend on the total energy, $\mathcal{E}$, and the mass density, $mn$. Assuming the form $R \propto \mathcal{E}^a \rho^b t^c$, a simple dimensional analysis immediately leads to the form [16]

$$R \sim \left(\frac{\mathcal{E}t^2}{mn}\right)^{\frac{1}{3}} \,, \tag{6}$$

but unlike the blast problem, $\mathcal{E}$ now decays with time. In the $x > 0$ half-line, the total number of moving particles $\mathcal{N}_+(t) \sim nR \sim (\mathcal{E}/m)^{1/3}(nt)^{2/3}$, and the energy $\mathcal{E}(t) \sim mv^2\mathcal{N}_+$. Together they give $v \sim (\mathcal{E}/(mnt))^{1/3}$ and inserting this in Eq. (5) gives $d\mathcal{E}/dt \sim -\mathcal{E}/t$ leading finally to Eq. (4). The exponent $\beta$ is the precise numerical factor in the equation $d\mathcal{E}/dt = -\beta\mathcal{E}/t$. This heuristic argument supports an algebraic decay law, but it does not allow one to fix a numerical value of the exponent which we will do later. By putting Eq. (4) into Eq. (6), and from dimensional considerations we obtain

$$R = An^{-1}\tau^\delta \,, \qquad \delta \equiv \frac{2-\beta}{3} \,, \tag{7}$$

where we define the dimensionless time $\tau = nv_0 t$, and $A$ is a dimensionless constant depending only on the mass ratio $\mu = m/M$ and the mass arrangement. We will later show that, using our results in Sec. (5), one can argue for the form:

$$\lim_{t\to\infty} \frac{nR(t)}{(nv_0 t)^\delta} = \begin{cases} A(\mu) & \text{for arrangement (3a)}\,, \\ A(\mu)\left(\frac{2\mu}{1+\mu}\right)^\delta & \text{for arrangement (3b)}\,. \end{cases} \tag{8}$$

This dependence on the detailed arrangement is a sign of the highly non-hydrodynamic behavior of the system. Similarly, the dimensionally complete form of (4) is

$$\mathcal{E} \sim \mathcal{E}_0 \tau^{-\beta} \,. \tag{9}$$

The total number of moving particles for $x > 0$ is $\mathcal{N}_+ \sim nR$, so from (7) we see that it scales algebraically, $\mathcal{N}_+ \sim \tau^\delta$. For the number, $\mathcal{N}_-(t)$, of particles in the initially empty half-line $x < 0$, we note that $\frac{d\mathcal{N}_-}{dt} \sim nv$, from which we get

$$\mathcal{N}_- \sim \mathcal{N}_+ \sim \tau^\delta \,. \tag{10a}$$

Thus the total number $\mathcal{N}_-$ of the particles in the initially empty half-line $x < 0$ and the total number $\mathcal{N}_+$ of the moving particles in half-line $x > 0$ exhibit the same scaling. The visual impression from the splash pattern (see Fig. 2) is that $\mathcal{N}_- \ll \mathcal{N}_+$. In spite of their significant disparity, we have verified numerically [see Sec. (4)] that both quantities exhibit identical scaling and their ratio approaches a small positive value

$$\lim_{t\to\infty} \frac{\mathcal{N}_-(t)}{\mathcal{N}_+(t)} = \lambda(\mu), \tag{10b}$$

that only depends on the mass ratio $\mu = m/M$. We find a dependence of $\mathcal{N}_\pm$ on the mass arrangement similar to what is seen in Eq. (8) for $R(t)$, however the above ratio does not have this dependence at long times.

To estimate the total number of collisions $\mathcal{C}(t)$ we notice that it grows according to the rate equation $d\mathcal{C}/dt \sim \mathcal{N}_+/\tau$, where $\tau$ is the mean collision time. We have $\tau^{-1} \sim nv \sim nv_0\tau^{-\frac{1+\beta}{3}}$, hence $d\mathcal{C}/d\tau \sim \tau^{\frac{1-2\beta}{3}}$ leading finally to

$$\mathcal{C} \sim \tau^{2\delta}. \tag{11}$$

The heuristic arguments support an algebraic decay law for the energy and the time-dependence of various other physical observables are all expressed in terms of a single exponent $\beta$. We also gain physical understanding of the main features of the evolution. To find the numerical value of the exponent, we will now proceed with a solution of the Euler equations in the hydrodynamic region.

## 3 Scaling solution of the second kind

As seen from the discussion in the previous section, the hydrodynamic region consists of a single shock front at the position $R(t) \sim n^{-1}\tau^\delta$. We expect that the hydrodynamic variables acquire the scaling form in the scaling region

$$t \to \infty, \quad |x| \to \infty, \quad \xi = x/R(t) < 1. \tag{12}$$

The mass density $\rho(x,t) = (m+M)n(x,t)/2$ that generally depends on the two variables, $x$ and $t$, becomes a function of a single scaling variable

$$\rho(x,t) = \rho_\infty G(\xi), \tag{13}$$

where $\rho_\infty = (m+M)n/2$ is the mean density of the initial undisturbed gas and $G(\xi)$ a scaling function.

To determine the behavior of the hydrodynamic variables behind the shock we first notice that the shock moves with velocity

$$U = \frac{dR}{dt} = \delta\frac{R}{t}. \tag{14}$$

Hence it is convenient to choose

$$v(x,t) = \delta\frac{R}{t}V(\xi), \tag{15}$$

as the scaling form of the velocity of the flow field behind the shock wave, $\xi < 1$, with $V(\xi)$ as the scaling function. In the blast problem, it is customary [32,33] to use the square of the speed of sound, $c^2$, instead of pressure (or temperature). For our one-dimensional hard-point gas $c^2 = 3p/\rho = 3T$ which gives us the scaling form:

$$p(x,t) = \frac{\rho c^2}{3} = \rho\frac{\delta^2}{3}\frac{R^2}{t^2}Z(\xi), \tag{16}$$

where $Z(\xi)$ is a scaling function. For the one-dimensional ideal gas, the velocity $v(x,t)$, mass density $\rho(x,t)$ and pressure $p(x,t)$ satisfy the Euler equations [32,33]

$$\partial_t\rho + \partial_x(\rho v) = 0, \tag{17a}$$

$$(\partial_t + v\partial_x)\ln(p/\rho^3) = 0, \tag{17b}$$

$$(\partial_t + v\partial_x)v = -\rho^{-1}\partial_x p, \tag{17c}$$

behind the shock wave, $x < R(t)$. The Rankine-Hugoniot relations [32] describing the jump between the hydrodynamic variables on both sides of the shock wave have a simple form

$$\frac{p(R)}{\rho_\infty U^2} = \frac{1}{2}, \quad \frac{\rho(R)}{\rho_\infty} = 2, \quad \frac{v(R)}{U} = \frac{1}{2}, \tag{18}$$

in our case when the pressure and temperature in front of the shock wave are equal to zero. Using the scaling forms (13), (15), (16) we re-cast the Rankine-Hugoniot relations (18) into boundary conditions for the scaling functions $(V, G, Z)$:

$$V(1) = \tfrac{1}{2}, \quad G(1) = 2, \quad Z(1) = \tfrac{3}{4}. \tag{19}$$

We substitute the scaling forms (13), (15), (16) into the Euler equations and arrive at three coupled ODEs for the scaling functions $(V, G, Z)$:

$$(GV)' = \xi G', \tag{20a}$$

$$(V - \xi)[\ln(Z/G^2)]' = 2\Delta, \tag{20b}$$

$$(V - \xi)V' + \frac{(GZ)'}{3G} = \Delta V, \tag{20c}$$

where $(\cdots)' = d(\cdots)/d\xi$ and we defined

$$\Delta = \frac{1 + \beta}{3\delta} = \frac{1 + \beta}{2 - \beta}. \tag{21}$$

In addition to the boundary conditions on the shock, Eq. (19), we need boundary conditions at $\xi \to -\infty$:

$$V(-\infty) = -\infty, \quad G(-\infty) = 0, \quad Z(-\infty) = 0. \tag{22}$$

These boundary conditions are natural from the requirement that the hydrodynamic region should smoothly connect, at the left end, with the non-hydrodynamic splatter region which consist of a low-density gas on ballistically moving non-interacting particles.

The challenge now is to solve Eqs. (20) subject to Eqs. (19) and (22). For the blast problem, the condition of conservation of energy allows us to determine exactly the exponent $\delta$ and the constant $\alpha$, and also a closed form solution for the scaling fields [16,32,33] — this is an example of a self-similar solution of the first kind [19]. In the "one-sided" splash problem, energy is not conserved in the full hydrodynamic regime as it is lost to the splatter particles [3,4]. In this case it turns out that one has to treat the system Eqs. (20,19,22) as a non-linear eigenvalue equation. Only for a unique choice of $\Delta$ (and therefore $\delta$) are the boundary conditions satisfied — this type of solutions are known in the literature as self-similar solutions of the second kind [19], first discovered by Guderley [21].

In order to proceed further, a slightly different formulation is useful. Following Ref. [32], we rewrite Eq. (20) as

$$G'(\xi) = -\frac{\Delta G \,(3V^2 - 3\xi V - 2Z)}{3(V - \xi)[(V - \xi)^2 - Z]}, \tag{23a}$$

$$V'(\xi) = \frac{\Delta \,(3V^2 - 3\xi V - 2Z)}{3[(V - \xi)^2 - Z]}, \tag{23b}$$

$$Z'(\xi) = \frac{2\Delta Z \,(3\xi V - 3\xi^2 + Z)}{3(V - \xi)[(V - \xi)^2 - Z]}. \tag{23c}$$

We then define new scaling functions $U$, $C$ through $V = \xi U$, $Z = \xi^2 C^2$ which leads to

$$\xi G'(\xi) = \frac{\Delta G \left(3U(U-1) - 2C^2\right)}{3(U-1)\left[C^2 - (U-1)^2\right]}, \tag{24a}$$

$$\xi U'(\xi) = \frac{C^2(2\Delta - 3U) - 3U(U-1)(1 + \Delta - U)}{3\left[C^2 - (U-1)^2\right]}, \tag{24b}$$

$$\xi C'(\xi) = \frac{C\left[C^2(3 + \Delta - 3U) + 3(U-1)\{\Delta + (U-1)^2\}\right]}{3(U-1)\left[C^2 - (U-1)^2\right]}. \tag{24c}$$

From Eqs. (24b,24c) we get

$$\frac{dC}{dU} = \frac{C\left[C^2(3 + \Delta - 3U) + 3(U-1)\{\Delta + (U-1)^2\}\right]}{(U-1)\left[C^2(2\Delta - 3U) - 3U(U-1)(1 + \Delta - U)\right]}. \tag{25}$$

At the shock we have $C = \sqrt{3}/2$ at $U = 1/2$. The condition that the numerators and denominators in Eqs. (24b,24c) vanish at the same point (so that $U, C$ are single valued functions of $\xi$), we get $C = 3$ at $U = -2$. Numerically we find that the solution of Eq. (25) satisfies the boundary conditions for a unique value of $\Delta$ which gives

$$\beta = 0.11614383675\ldots. \tag{26}$$

As a self-consistency check we have verified that, with this value of $\Delta = 0.592478268\ldots$, the solution of Eqs. (20) satisfies the boundary conditions in Eqs. (19) and Eqs. (22). Unlike the TvNS solution of the blast problem, in the present case, we are not able to determine exactly the dimensionless constant $A$ in Eq. (7), and also we do not have an exact solution of the scaling functions. Instead, we obtain $A$ numerically by choosing it in such a way that the boundary conditions are satisfied. In Sec. (4) we will provide numerical evidence for the above value of the exponent $\beta$ and the numerically obtained scaling functions.

**Comparisons with the vWZ problem**: The macroscopic part, $\xi = O(1)$, of the splash problem in one dimension resembles the impulsive loading problem studied by von Weizsäcker [34] and Zeldovich [35]; see [19, 36] for textbook accounts. We refer to this as the vWZ problem. In the vWZ setting, the gas at zero pressure and temperature occupies the half-space $x \geq 0$; the pressure is suddenly created at $x = 0$ at time $t = 0$ and removed at $t = t_r$.

The advantage of the splash problem is that one does not need an extra parameter $t_r$, it suffices to suddenly hit the left-most particle. Even more significant virtue is the one-dimensional nature of the splash problem allowing direct molecular dynamic simulations. The emerging results can be compared with continuous predictions. We emphasize that the intriguing aspects of the splash problem concerning the freezing [see Sec. 5] are non-hydrodynamic, they cannot be accounted for by continuous treatment.

At first sight, our continuous treatment looks identical to the analysis [19, 34–36] of the vWZ problem. There is a caveat, however, which we now demonstrate by computing the total momentum and the energy of the system. The momentum $\mathcal{P} = \int dx\, \rho(x, t)v(x, t)$ reduces to

$$\mathcal{P} = \rho_\infty \delta \frac{R^2}{t} \int_{-\infty}^1 d\xi\, G(\xi)V(\xi), \tag{27}$$

while the energy $\mathcal{E} = \int dx\, \rho\left[\frac{v^2}{2} + \frac{T}{2}\right]$ becomes

$$\mathcal{E} = \rho_\infty \delta^2 \frac{R^3}{6t^2} \int_{-\infty}^1 d\xi\, G(\xi)\left[3V^2(\xi) + Z(\xi)\right]. \tag{28}$$

Since $R^2/t \sim t^{(1-2\beta)/3}$ diverges as $t \to \infty$, while the total momentum is finite, so Eq. (27) seemingly gives

$$\int_{-\infty}^1 d\xi\, G(\xi)V(\xi) = 0. \tag{29}$$

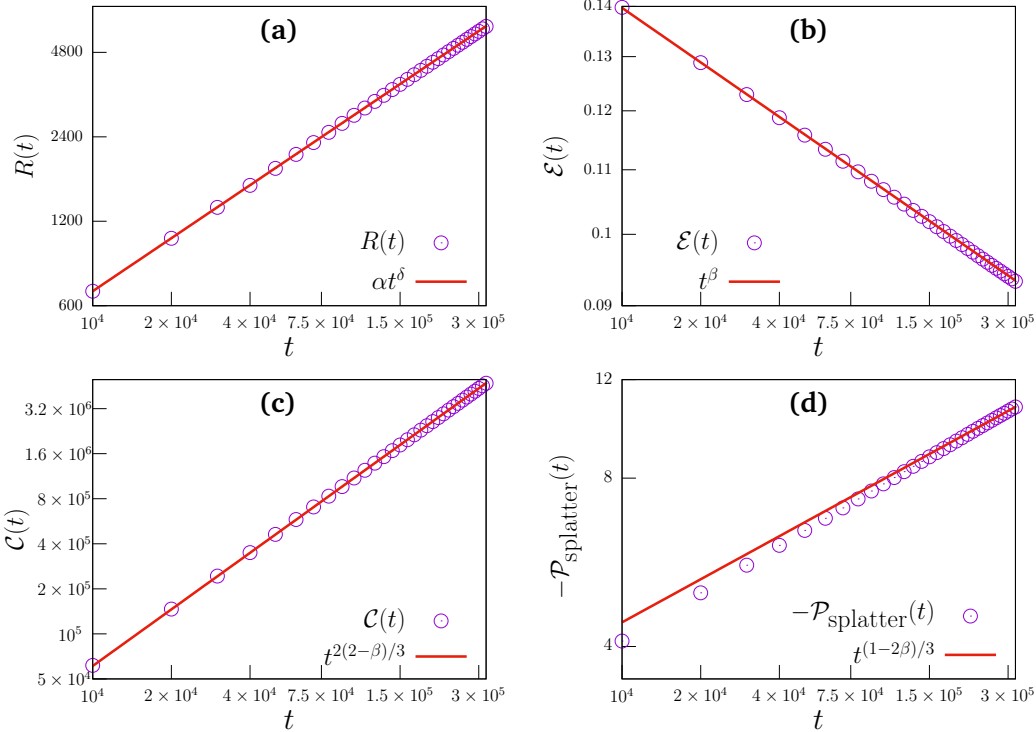

Figure 3: Comparison of microscopic simulation results (points) with analytic predictions (solid lines) for the time-dependence of various observables: (a) The shock position $R(t)$, evaluated from simulations as the average position of the rightmost moving particle at time $t$. The slope of the solid line $R(t) = \alpha t^{\delta}$, was found to be $\alpha = 2.08$; (b) total energy of the particles $\mathcal{E}(t)$ in $x > 0$ region; (c) total number of collisions $\mathcal{C}(t)$; (d) total momentum $\mathcal{P}_{\text{splatter}}(t)$ in the $x < 0$ region. The particles are distributed uniformly with density $\rho_{\infty} = 1$, $v_0 = 1$, and the results are all shown for the mass arrangement (3a), with $m = 2/3, M = 4/3$. Averages over $2 \times 10^5$ realizations were done. In all cases the value of $\beta$ given by Eq. (26) was used and $\delta = (2 - \beta)/3$.

Similarly $R^3/t^2 \sim t^{-\beta}$ vanishes as $t \to \infty$, while the total energy is finite, so Eq. (28) implies

$$\int_{-\infty}^{1} d\xi\, G(\xi)\big[3V^2(\xi) + Z(\xi)\big] = \infty\,. \tag{30}$$

Integral relations (29)–(30) indeed appear in the analysis of the vWZ problem, see [19]. In the realm of the splash problem, however, Eqs. (27) and (28) give the momentum and energy of the continuous part only, the full momentum and energy also contain the contribution from the splatter particles. The energy of the continuous part indeed decays, $\mathcal{E} \sim t^{-\beta}$, so the integral in (30) remains finite. The unbounded growth of $\mathcal{P}$ is also not a problem, it is compensated by the momentum of the splatter particles scaling as $\mathcal{P}_{\text{splatter}} \sim -t^{(1-2\beta)/3}$, see Eq. (39). Thus the integral in (29) also remains finite. Summarizing

$$\int_{-\infty}^{1} d\xi\, GV = I_1\,, \quad \int_{-\infty}^{1} d\xi\, G\big[3V^2 + Z\big] = I_2\,, \tag{31}$$

with $0 < I_1 < \infty$ and $0 < I_2 < \infty$.

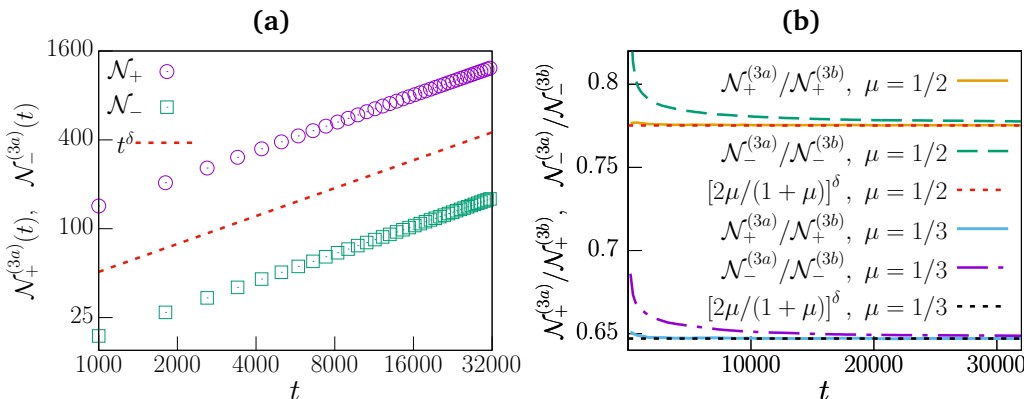

Figure 4: Panel (a): Comparison of microscopic simulation results with analytic predictions for the time-dependence of the total number of excited particles $\mathcal{N}_-(t)$ in the $x < 0$ region and the number $\mathcal{N}_+(t)$ in the region $x > 0$ for the mass arrangement (3a). The particles are distributed uniformly with density $\rho_\infty = 1$, $v_0 = 1$, and $m = 2/3$, $M = 4/3$. Panel (b): Here we test the prediction, Eq. (8), that the ratio of the values of $\mathcal{N}_+ \sim nR(t)$ (and $\mathcal{N}_-$), obtained from the two arrangements (3a) and (3b) is given, at long times, by the $\mu$-dependent factor $2\mu/(1+\mu)$. In all cases, averages are taken over $2 \times 10^5$ realizations. The value of $\beta$ given by Eq. (26) was used, with $\delta = (2-\beta)/3$.

## 4 Numerical results

We now compare the analytic predictions obtained in Secs. (2,3), with the results obtained from direct microscopic simulations of the AHP model described in Sec. (2). Since the dynamics consists only of free evolution and elastic collisions, this can be simulated very efficiently using an event-driven algorithm. We mostly present results for the mass arrangement (3a), with mass density $\rho_\infty = 1$, masses $m = 2/3$ and $M = 4/3$, and initial velocity $v_0 = 1$. We have done computations with two other mass ratios ($\mu = 1/3$, $1/10$) and verified that the main results do not change, except for the values of some prefactors.

In Fig. (3) we show the comparisons of results from microscopic simulations and the analytic predictions for the following quantities: (a) position $R(t)$ of the right most moving particle; (b) the total energy of the particles $\mathcal{E}(t)$ in $x > 0$ regime at different times; (c) the total number of collisions $\mathcal{C}(t)$ experienced by all the particles; (d) the total momentum $\mathcal{P}_-(t)$ of the splatter particles. In all cases, we took averages over $2 \times 10^5$ realizations chosen from an ensemble of initial conditions with particles uniformly distributed on the half line $x > 0$ and with mass density fixed at $\rho_\infty = 1$. We chose configuration (3a) and $v_0 = 1$. As can be seen, we find excellent agreement with the analytic predictions with the value of $\beta = 0.116143836....$ In particular we find agreement with Eq. (7) for $\alpha = 2.08$. In Fig. (4)(a), we plot the total number of moving particles, $\mathcal{N}_-(t)$ and $\mathcal{N}_+(t)$, in the regions $x < 0$ and $x > 0$ respectively, and verify the asymptotic growth law predicted in Eq. (10a). The validity of Eq. (10b), in particular the dependence of the ratio on the mass ratio $\mu$, is also clear from the plots in Fig. (4)(b).

Next we consider the scaling form of the three conserved fields. The three fields for density $\rho(x,t)$, velocity $v(x,t)$ and energy $E(x,t)$ can be obtained from the microscopic simulations

using the basic definitions:

$$\rho(x,t) = \sum_{j=0}^{\infty} \left\langle m_j \, \delta[q_j(t) - x] \right\rangle, \tag{32a}$$

$$\rho(x,t)v(x,t) = \sum_{j=0}^{\infty} \left\langle m_j v_j \, \delta[q_j(t) - x] \right\rangle, \tag{32b}$$

$$E(x,t) = \sum_{j=0}^{\infty} \frac{1}{2} \left\langle m_j v_j^2 \, \delta[q_j(t) - x] \right\rangle, \tag{32c}$$

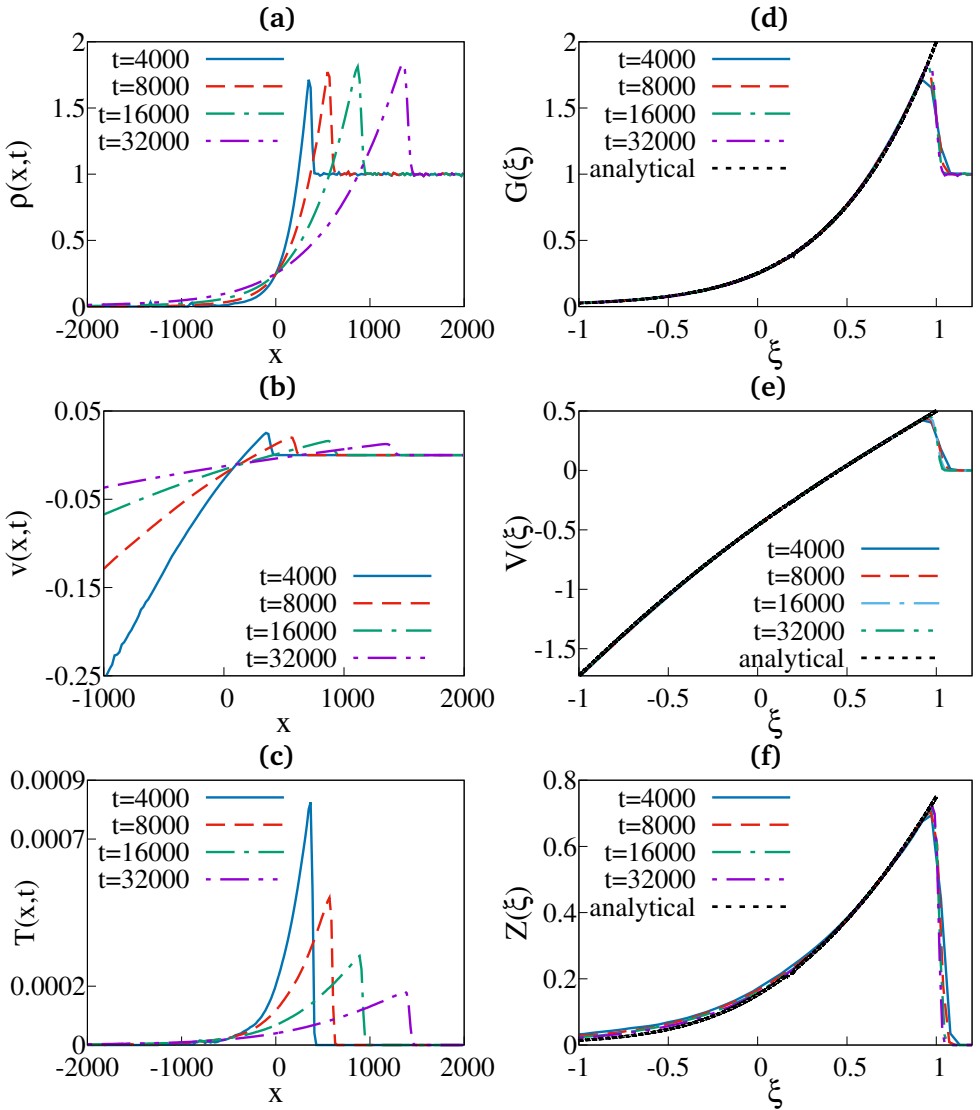

Figure 5: Comparison of microscopic simulation results with analytic predictions for the scaling functions corresponding to the hydrodynamic conserved fields: (a),(b),(c) show plots of the profiles of density, velocity and temperature fields calculated from molecular dynamic simulations, at different times. The simulation parameters were $\rho_\infty = 1$, $v_0(0) = 1$, $m = 2/3$, $M = 4/3$, with the mass arrangement (3a), and an ensemble average over $10^5$ initial conditions were performed. In (d),(e),(f) we see the scaling collapse of the simulation data and we see an excellent agreement with the analytic scaling functions $G(\xi)$, $V(\xi)$ and $Z(\xi)$ (black dashed line). The scaling variable was taken as $\xi = x/R(t)$ with $R(t) = 2.08 t^{\delta}$.

where $\langle ... \rangle$ indicates an average over the uniform positional distribution. The temperature field is then given by $T(x,t) = (m+M)(E/\rho - v^2)/2$. In Figs. 5(a)(b)(c), we plot $\rho(x,t)$, $v(x,t)$ and $T(x,t)$ at different times. We clearly observe the shock front in all the fields. In Figs. 5(d)(e)(f), we find that using the scaling variable $\xi = x/R(t)$ with $R = \alpha t^\delta$, $\alpha = 2.08$ gives us a very good scaling collapse. The analytic scaling functions $G, V, Z$, obtained numerically from Eqs. (23) are shown by black dashed lines in Figs. 5(d)(e)(f), and are seen to agree remarkably well with the simulation results.

## 5 Extreme particles in the splatter

We now examine the structure of the splatter, which is a highly non-hydrodynamic region. The right-most moving particle $X_+(t) = R(t)$ advances sub-ballistically, namely as $\tau^\delta$. In contrast, the left-most particle propagates ballistically into an initially empty half-line:

$$\lim_{t\to\infty} \frac{X_-(t)}{v_0 t} = \begin{cases} w_0 & \text{for arrangement (3a)}, \\ W_0 & \text{for arrangement (3b)}. \end{cases} \tag{33}$$

The reflection velocities $w_0$ and $W_0$ are dimensionless and depend only on the mass ratio $\mu = m/M$ and the type of the initial arrangement, and in general on the details of particle positions.

The particles are labeled $0, 1, 2, \ldots$, so the left-most particle has label 0. The initial velocities are $v_j = \delta_{j,0}$; we set $v_0 = 1$ without loss of generality. After a few collisions, the left-most particle acquires a certain ultimate velocity and propagates into the half-line $x < 0$ without experiencing further collisions. The same fate is shared by other particles: Every particle eventually ceases to collide and propagates to the left with a certain ultimate velocity. For the $i^{\text{th}}$ particle, we denote by $v_0 w_i$ [resp. $v_0 W_i$] the ultimate velocity for arrangement (3a) [resp. (3b)]; the dependence on $v_0$ is trivial, so we focus on the dimensionless velocities $w_i$ and $W_i$. We also denote by $c_i$ [resp. $C_i$] the number of collisions the $i^{\text{th}}$ particle has experienced. All ultimate velocities are negative: $w_i < 0$ and $W_i < 0$ for all $i = 0, 1, 2, \ldots$.

The above description of the fate of the system is intuitively appealing, but not proven. We now propose two conjectures about our infinite-particle billiard:

**C1** For each particle, collisions eventually cease and the particle then moves forever with a fixed ultimate velocity.

**C2** The left-most particle experiences only one collision for arrangement (3a).

These conjectures seem reasonable and we can verify them in our numerical simulations, for a large number of random initial configurations (of particle positions). A general rigorous proof could be a very tough challenge. For a *finite* system of hard spheres in $\mathbb{R}^d$, a similar freezing condition like **C1** implies that the total number of elastic collisions is always finite. This was guessed by Sinai [37] who proved it in one dimension. In an arbitrary dimension, the proof was found in [38], see [39,40] for other proofs. The maximal number of collisions between three identical hard spheres is four [41,42]; for four or more spheres, the answer is unknown. When the number of spheres is large, the total number of collisions can be very large; finding a good upper bound is an active area of research [43–47]. The number of spheres is infinite in the splash problem and hence the total number of collisions grows without a bound, c.f. Eq. (11) in one dimension. Yet each sphere eventually escapes into the half-space $x < 0$ and collisions cease. Thus **C1** must be true in an arbitrary dimension. The one-dimensional case is the simplest and proving **C1** may be feasible. Finally, the conjecture **C2** is intuitively very plausible, but we are able to prove it only for the case $\mu < 1/3$ (see below).

We now probe the ultimate characteristics of the extreme particles. We begin with arrangement (3a) and consider a specific random configuration $S = \{q_0 = z_0, q_1 = z_1, q_2 = z_2, \ldots\}$. After the first collision involving the left-most particle and particle 1, the post-collision velocities are

$$v_0^{(1)} = \frac{\mu - 1}{\mu + 1}, \quad v_1^{(1)} = \frac{2\mu}{\mu + 1}. \tag{34}$$

(The notation $v_i^{(j)}$ refers to the velocity of the $i^{\text{th}}$ particle after it has experienced $j$ collisions.) We note that the condition, $-v_0^{(1)} > v_1^{(1)}$, is obtained for $\mu < 1/3$, and implies that the second particle can never collide again with the first particle. This thus proves **C2** for $\mu < 1/3$.

For arrangement (3a), **C2** implies that

$$w_0 = v_0^{(1)} = \frac{\mu - 1}{\mu + 1}, \qquad c_0 = 1. \tag{35a}$$

Particles labeled $1, 2, 3, \ldots$ in arrangement (3a) constitute a specific configuration $S' = \{q_0 = z_1, q_1 = z_2, q_2 = z_3, \ldots\}$, with mass arrangement (3b). Now, according **C2**, the left-most particle ceases to collide after the first collision that has led to (34); we should only keep in mind that particle 1 initially moves with velocity $v_1^{(1)} = \frac{2\mu}{\mu+1} < 1$ while other particles are at rest. Hence it follows

$$w_1(S) = \frac{2\mu}{\mu + 1} W_0(S'), \qquad c_1(S) = C_0(S') + 1, \tag{35b}$$

where we have explicitly mentioned the arguments $S$ and $S'$ to emphasize that the results are true for these two specific random positional configurations. Similarly, for the following particles, we get

$$w_{k+1}(S) = \frac{2\mu}{\mu + 1} W_k(S'), \qquad c_{k+1}(S) = C_k(S'), \tag{35c}$$

for $k \geq 1$. A little discrepancy between (35b) and (35c) is due to the fact that particle 1 has experienced the collision with the left-most particle and this distinguish it from the left-most particle in arrangement (3b). The above equations are not true for $w_k$ and $W_k$ (or $c_k$, $C_k$) evaluated for two *independently* chosen random position configurations. In Fig. (6)(a-d) we verify that indeed Eq. (35c) is true only when $S'$ is chosen in a specific way. However, if we now average over the uniform distribution of initial positions, then we get

$$\langle w_{k+1} \rangle = \frac{2\mu}{\mu + 1} \langle W_k \rangle, \qquad \langle c_{k+1} \rangle = \langle C_k \rangle + \delta_{k,0}. \tag{36}$$

Therefore it suffices to determine $\langle W_k \rangle$ and $\langle C_k \rangle$ characterizing arrangement (3b) for all $k \geq 0$; the corresponding quantities $\langle w_k \rangle$ and $\langle c_k \rangle$ characterizing arrangement (3a) are then found from (36).

Energy conservation for arrangement (3a) gives

$$\mu \sum_{i \geq 0} w_{2i}^2 + \sum_{i \geq 0} w_{2i+1}^2 = \mu, \tag{37a}$$

while energy conservation for arrangement (3b) leads to

$$\sum_{i \geq 0} W_{2i}^2 + \mu \sum_{i \geq 0} W_{2i+1}^2 = 1. \tag{37b}$$

Only one of these sum rules is independent. Indeed, using Eqs. (35a)–(35c) one can recast (37a) into (37b).

The quantities $W_k$ and $C_k$ exhibit simple behaviors

$$W_k \sim -k^{-\frac{1+\beta}{2-\beta}}, \qquad C_k \sim k, \tag{38}$$

when $k \gg 1$. Indeed, using the estimates for the typical velocity $v \sim \tau^{-(1+\beta)/3}$ and the typical number of collisions per particle $C \sim \mathcal{C}/R \sim \tau^{(2-\beta)/3}$, and expressing the label through time $k \sim R \sim \tau^{(2-\beta)/3}$, one gets (38). In Fig. (6)(e,f) we show the numerical verification of Eqs. (36,38).

As a consistency check we note that the sums in (37b) converge since $\frac{1+\beta}{2-\beta} > \frac{1}{2}$. Using (38) we can estimate the momentum of the splatter

$$\mathcal{P}_{\text{splatter}} = \sum_{j=0}^{k} W_j \sim -k^{\frac{1-2\beta}{2-\beta}} \sim -\tau^{\frac{1-2\beta}{3}}, \tag{39}$$

which agrees with our earlier estimate of the splatter momentum using the hydrodynamics picture [see discussion before Eq. (31)].

Finally we note that, following arguments as those leading to Eq. (35b), lead us to equations, such as Eq. (8), relating the long time asymptotic forms of $R(t), \mathcal{N}_{\pm}(t)$ for the two arrangements (3a) and (3b).

# 6  Conclusions

In the 1D blast problem studied in [16,17], the spreading of an intense localized energy burst in an infinite cold gas was investigated and it was shown that at long times, a self-similar scaling solution emerges. This solution could be accurately described by continuum hydrodynamics. Here we studied a related problem, namely the 'one-sided' blast problem, viz. the splash problem, where a semi-infinite cold gas on the positive half line is excited at the origin. The splash problem differs significantly from the blast problem and we summarize here some of the striking results that we find:

- A shock front develops in the gas, with position given by $R(t) = An^{-1}(nv_0 t)^{\delta}$. The hydrodynamic fields behind the shock have a self-similar form at long times. The exponent $\delta$ and the form of the scaling functions differ from the blast problem. This difference is due to the fact that the energy of the gas in the hydrodynamic region decays slowly with time. This leads us to self-similar solutions of the second kind, where the exponent cannot be fixed from dimensional arguments alone. From a numerical solution of a nonlinear eigenvalue equation we determine $\delta = (2 - \beta)/3$, with $\beta = 0.11161438....$ Thus $\delta$ is reduced from its value $2/3$ for the blast problem. We find excellent agreement between our numerically evaluated analytic scaling functions and results of direct molecular dynamics simulations. Note that unlike the blast problem [16,17], here we do not have a core region with high gradients of the fields so that dissipative terms become important. Thus the Euler equations are sufficient.

- Coexisting with the hydrodynamic region is the splatter, formed by recoiled particles that are contained in the region $x < 0$. The splatter particles exhibit decisively non-hydrodynamic behavior and travel ballistically with negative velocities. At long times, most of the total injected energy is contained in the splatter. The total negative momentum carried by the splatter increases with time as $t^{\frac{1-2\beta}{3}}$.

- We conjecture that the velocities of an increasing number of particles at the left end of the splatter get frozen after a finite number of collisions. The precise values of the frozen

velocities depend on the mass arrangement as well as the initial positions of the particles. For the mass arrangement (3a) we further conjecture that the left most particle suffers only one collision and always moves with an eventual velocity $w_0 = (1-\mu)/(1+\mu)$. The remaining $k$ frozen particles form a fan with velocities $w_0 < w_1 < w_2$.... for arrangement (3a) and $W_0 < W_1 < W_2$.... for arrangement (3b). We find exact relations between these two sets when they are averaged over the random initial positional configurations.

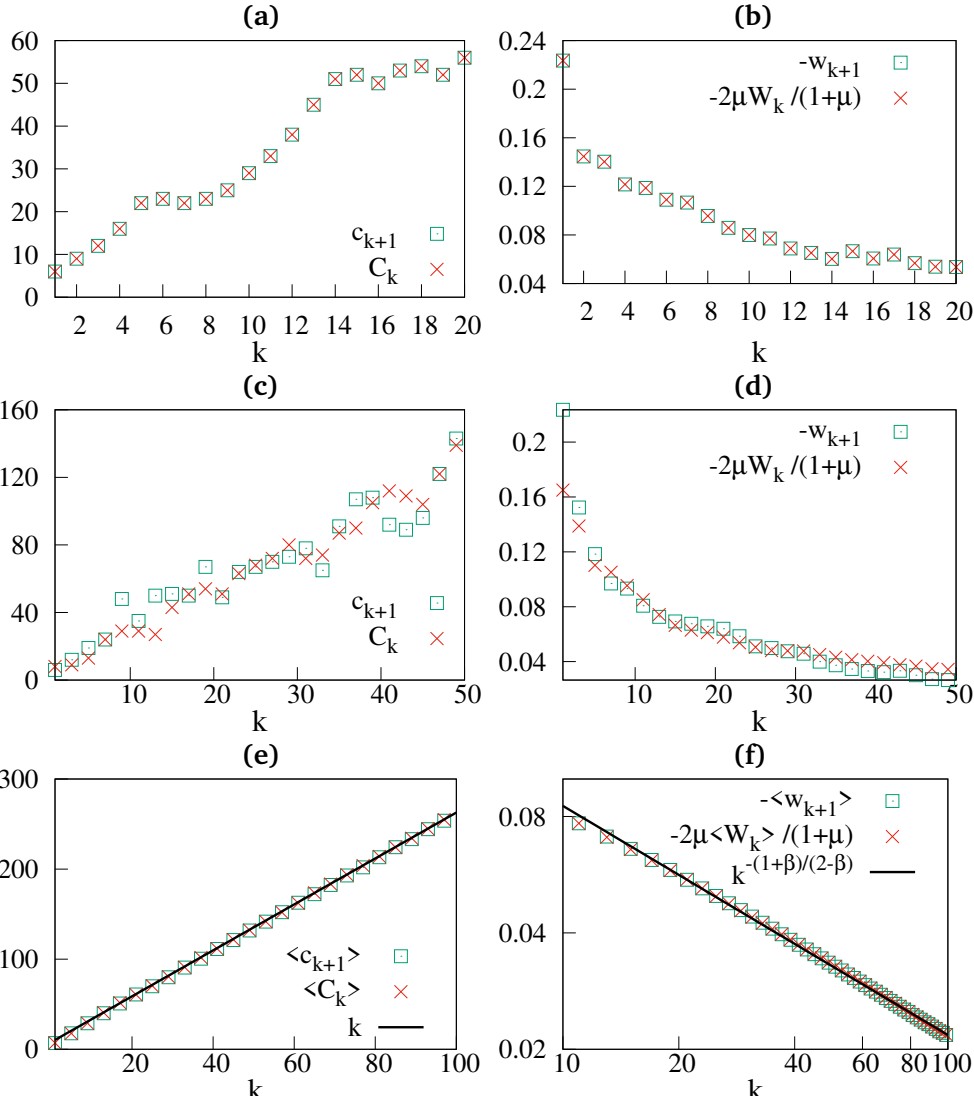

Figure 6: (a,b) Verification of Eqs. (35c) from microscopic simulations. We computed the number of collisions experienced and the frozen velocities of different splatter particles after a sufficiently long time ($t = 6.4 \times 10^5$). In all cases we checked that increasing time by a factor of 10 did not lead to any more collisions for these particles. In this case, we considered a single random initial configuration $S$, while $S'$ was obtained by deletion of the first particle coordinate. (c,d) In this case, two completely independent and random realizations $S$ and $S'$ are chosen and in this case, we see that Eq. (35c) no longer holds. In (e,f) we show the results where we average over $10^3$ configurations drawn from a uniform distribution at mean density $n = 1$. We again find that the equalities in Eq.(36) hold. The black lines are the asymptotic estimates from Eq. (38).

- Using the results of the hydrodynamic regime, and using heuristic arguments we are able to make several predictions for the long time properties in the splatter. For example we predict that the mean number of collisions of the $k^{\text{th}}$ particle from the left end, before it gets frozen, scales linearly with $k$ while the mean frozen velocity $W_k \sim -k^{\frac{1+\beta}{2-\beta}}$.

Proving some of the above conjectures and an exact determination of the exponent $\beta$ are some interesting future problems. A non-trivial feature of our work is that we are able to make detailed predictions for the non-hydrodynamic region by making use of the hydrodynamic scaling solution. It would be interesting to extend these results to more realistic physical problems such as splashes in higher dimensions.

## Acknowledgments

We acknowledge the ICTS program on "Hydrodynamics and fluctuations - microscopic approaches in condensed matter systems (ONLINE)" (Code: ICTS/hydro2021/9)" for enabling crucial discussions related to this work. AD and SC acknowledge support of the Department of Atomic Energy, Government of India, under Project No. 19P1112R&D.

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
