# Peer review of "A splash in a one-dimensional cold gas"

_SciPost Physics, doi:SciPost Phys. 13, 074 (2022)_

## Round 1 · Referee Report · Anonymous (Referee 1) · 2022-5-10

Strengths

1) The work is well done and based on analytic calculations
2) The phenomenon described in the paper is intriguing (non-intuitive)

Weaknesses

The main and only weakness I see is the peculiarity of the problem: perhaps too special a setup to draw general conclusions

Report

I am unsure about the suitability in this journal, because the authors deal with a very special initial condition.
If the authors are able to justify the general relevance of their study (showing for instance some form of robustness), my doubts would disappear.

Requested changes

1) Are the authors able to show that this scenario is in some sense robust?

2) While studying heat conductivity in the diatomic hard point gas, some past papers suggested that the special values of the mass ratio (special beyond the equal-mass case) might display a qualitatively different behavior.
In this work the mass ratio seems to be an ``innocent parameter" playing no special role. I would like to read a bit more on its role.

3) It is not entirely clear the way the initial condition is selected. Except for the leftmost particle, all are at rest, but what about the initial position?
Perhaps I have missed the point where this point is mentioned; however, even if I am wrong, it should be discussed more extensively, especially to comment about potential differences.

4) When the parameter beta is first mentioned, it would be useful to briefly anticipate how small it is to help the reader intuition.

5) When the function A(mu) is first mentioned, it would be useful to anticipate the expected dependence (this point is connected to point 2).

6) In Eq. (14), it would be useful to make it more explicit that it is a definition of V.

7) It is not clear how Eq. (15) comes out and what is the meaning of Z.

8) Coming to the numerical part, the agreement is always very good except for panel 3d. The authors should add some comments especially
on the amplitude of the deviations.

9) Finally, about the splatter particles. I understand that it is utterly difficult to prove rigorous statements. In fact, the authors limit themselves to proposing conjectures.
I am perfectly fine, but I have a curiosity: do they have any idea (perhaps just based on numerics) on the position of the last collision of the particles that progressively enter this region?

  • validity: high
  • significance: high
  • originality: high
  • clarity: good
  • formatting: excellent
  • grammar: excellent

Author:  Subhadip Chakraborti  on 2022-07-15  [id 2662]

(in reply to Report 1 on 2022-05-10)

We thank the referee for constructive criticisms, to which we provide a response below in the attached file. We are submitting a revised manuscript where we have made some changes in response to the comments of the referees. All changes are marked in blue in the revised manuscript (already resubmitted).

Attachment:

ResponseV3_Q1dk3h3.pdf

---

## Round 1 · Referee Report · Anonymous (Referee 2) · 2022-5-20

Strengths

1- I find the results reported in this paper very interesting and relevant for a broad community of physicists interested in the transport properties of low dimensional systems.

2- The paper is well-written, all results are explained first at the heuristic level and then supported with hydrodynamic solutions, together with numerical simulations confirming the overall picture.

3- Moreover, some of the results reported are intriguing, as e.g. the fact that the splatter asymptotically concentrates most of the energy in the system, with the energy of particles in the positive half-line (x>0) decaying algebraically in time. The presence of a hydrodynamic, sub-ballistic shock front coexisting with a non-hydrodynamic ballistic splatter also seems intriguing.

Weaknesses

1- I miss a discussion on the role of dissipative effects in the splash problem.

2- I also miss some discussion on how the anomalous transport properties of this model may affect the results described in this paper.

Report

In this paper, Chakraborti, Dhar and Krapivsky study the splash problem in a one-dimensional cold gas of point particles with binary (alternating) masses. In particular, they study the cascade of activity that results from the excitation of the leftmost particle with a fixed velocity in an otherwise frozen environment. They find (1) a sub-ballistic shock front propagating into the cold gas and separating it from a thermalized region, well-described by self-similar solutions to the Euler hydrodynamic equations, and (2) a non-hydrodynamic splash region formed by recoiled particles moving ballistically in the opposite direction. The authors offer heuristic derivations of the main scaling relations relevant for this problem, as well as self-similar solutions of the second kind to the Euler hydrodynamic equations governing the thermal part of the shock wave. They complement these analytical results with extensive molecular dynamics simulations of the microscopic model, finding excellent agreement with predictions in all cases.

In my opinion this is a very nice paper, as otherwise usual for these authors. I find the results reported in this paper very interesting and relevant for a broad community of physicists interested in the transport properties of low dimensional systems. The paper is well-written, all results are explained first at the heuristic level and then supported with hydrodynamic solutions, together with numerical simulations confirming the overall picture. Moreover, some of the results reported are intriguing, as e.g. the fact that the splatter asymptotically concentrates most of the energy in the system, with the energy of particles in the positive half-line (x>0) decaying algebraically in time. The presence of a hydrodynamic, sub-ballistic shock front coexisting with a non-hydrodynamic ballistic splatter also seems intriguing.

For these reasons, I recommend the publication of the present manuscript. I have however a number of comments and questions that may help to improve the manuscript:

* After reading in detail the paper, I keep wondering: Does dissipation play any role in the splash problem? In particular, do you expect thermal conduction and viscous dissipation effects to play a role in this problem? The authors mention the blast problem of Refs. [16,17] and its description in terms of the Taylor-von Neumann-Sedov (TvNS) self similar solution to the Euler equations. There it was shown that dissipative corrections were important near the blast core region. Can something similar happen in the splash problem? If so, what type of corrections do you expect in this case? A comment on this issue would be more than welcome.

* My next comment is directly related to the previos question. In particular, it is also well-known that transport is anomalous for this 1d particle system, with a heat conductivity that diverges as a power law of the system size. Does this anomalous transport properties affect in some way the results described in this paper?

* I’m curious about how hydrodynamic profiles relax behind the shock front, specially in the asymptotic far tail. In studies of shock wave propagation for the same model excited with a piston moving at constant velocity (see Ref. [14] in the paper), it was found that hydrodynamic profiles relax algebraically far from the shock front, in contrast with standard hydrodynamics predictions. Can something similar happen in the far tail of the hydrodynamic region for the splash problem? What are the predictions obtained from the scaling solution of the second kind to the Euler equations in this case? What is the observed asymptotic relaxation in the molecular dynamics simulation results? It would be interesting to include a comment on this issue in case the authors have enough data in this region.

* Mass ratio: In order to compare analytical predictions with molecular dynamics results, the authors choose particular values for the light and heavy particle masses, m=2/3 and M=4/3, which result in a mass ratio $\mu=1/2$. The value of this mass ratio affects relaxation timescales in the alternating hard point gas, with $\mu=1/2$ corresponding approximately to fastest relaxation. Is this the reason why the authors choose $\mu=1/2$? What happens for other mass ratios?

I have also a number of minor comments and questions:

* The exponent $\delta$ characterizing the growth of the hydrodynamic region in the splash problem takes a value $\delta=0.6279…,$ smaller than the equivalent exponent in the blast problem (2/3). Since both problems essentially differ in their symmetry properties, is there any symmetry argument supporting $\delta<2/3$?

* For the interested reader, and in order to make the paper more self-contained, it would be nice to have more details on the derivation of Eq. (5) using dimensional analysis (without having to resort to Ref. [16]))

* Typo: “bbut” right after Eq. (5).

* In the next paragraph below Eq. (21), third line, the authors mention constant $\alpha$ but I couldn’t find it in the main text. Similarly, in the second paragraph of section 4 (Numerical Results), a value of $\alpha=2.08$ in Eq. (6) is mentioned, but there is no $\alpha$ in Eq. (6). The caption of Fig. 3 suggests that $\alpha$ is just the amplitude of shock position with time. Defining $\alpha$ in the main text would probably clarify the discussion.

* In the line below Eq. (25), I would suggest to add the explicit numerical value of exponent $\Delta$ obtained from exponent $\beta$.

* Fig. 3: The agreement between the scaling predictions for the shock position, the total energy in the positive half-plane, and the total number of collisions, panels 3.a-c, is excellent for all times explored. However, the agreement for the total momentum in the splatter region (panel 3.d) is only asymptotically good for long enough times. Is there any particular reason for this discrepancy?

* Caption of Fig. 4, fourth line: The sentence containing “… that the ratio $N_+$ values obtained from …” seems difficult to understand.

* Fig. 5: Please make this figure bigger (maybe a panel with 2 columns and 3 rows would make it).

  • validity: high
  • significance: high
  • originality: high
  • clarity: high
  • formatting: excellent
  • grammar: excellent

Author:  Subhadip Chakraborti  on 2022-07-19  [id 2668]

(in reply to Report 2 on 2022-05-20)

We thank the referee for constructive criticisms, to which we provide a response below in the attached file. We are submitting a revised manuscript where we have made some changes in response to the comments of the referees. All changes are marked in blue in the revised manuscript (already resubmitted).

Attachment:

ResponseV4.pdf

---

## Round 2 · Referee Report · Anonymous (Referee 3) · 2022-7-19

Report

The authors have successfully addressed all the comments and questions raised in my previous report. The new version of the paper is now clearer and, as I said in my previous report, the results described here are very interesting and relevant for a broad community of physicists interested in the transport properties of low dimensional systems. For these reasons I recommend the publication of the current version of the manuscript in SciPost.

---

## Round 2 · Author Response

Dear Editor,
We thank both the referees for their constructive criticisms, to which we provide a response in the particular sections. We are
submitting a revised manuscript where we have made some changes in response to the comments of the referees. All
changes are marked in blue in the revised manuscript.
Regards,
Subhadip Chakraborti, Abhishek Dhar and P. L. Krapivsky

---

## Round 2 · List of Changes

All changes are marked in blue in the revised manuscript. Here is a list of major changes.

1) We have displayed Eq. (1) for better visibility.

2) We have added a new paragraph in the introduction and a sentence in the conclusion which describes the importance of this paper and explains why it is of more general relevance, and has implications beyond the specific case treated in this paper.

3) We have added few lines in Sec (2) to describe the interaction of the particles in detail.

4) We have explained the origin of Eq. (6) from dimensional analysis just before it.

5) We have explicitly mentioned that G, V and Z are three scaling functions.

6) We have modified Fig. (3) and plotted them on the same time range.

7) Modified the caption of Fig. (4).

8) Updated the arrangement of figures in Figs. (5) and (6) for better visibility.

9) We have added two new references.

10) We have polished the paper throughout to make it more clear and to response to some of the issues raised by the referees.

---

## Editorial Decision

published